# Time-Related Eating Patterns Are Associated with the Total Daily Intake of Calories and Macronutrients in Day and Night Shift Workers

**DOI:** 10.3390/nu14112202

**Published:** 2022-05-25

**Authors:** Catarina Mendes Silva, Bruno Simão Teixeira, Kenneth P. Wright, Yara Cristina de Paiva Maia, Cibele Aparecida Crispim

**Affiliations:** 1Graduate Program of Health Sciences, Faculty of Medicine, Federal University of Uberlândia, Uberlândia 38400-902, Brazil; catarinamsilva@yahoo.com.br (C.M.S.); brunosimao2005@hotmail.com (B.S.T.); yaracpmaia@gmail.com (Y.C.d.P.M.); 2Department of Integrative Physiology, University of Colorado Boulder, 3100 Marine Street, Boulder, CO 80309, USA; kenneth.wright@colorado.edu

**Keywords:** shift work, time-related eating patterns, mealtime, late eating intake, chrononutrition

## Abstract

The aim of the study was to investigate whether time-related eating patterns are associated with the daily intake of calories and macronutrients in Brazilian male military police officers (*n* = 81; 29-day and 52-night workers; mean age: 36.4 ± 0.9 and 38.5 ± 0.7 years, respectively). Energy and macronutrient intake were determined by a non-consecutive 3-day food recall. Time-related eating patterns, such as the time of the first and the last meals, eating duration, and caloric midpoint, were evaluated. Individuals were classified as “early” or “late” eaters according to the median caloric midpoint. Night shift workers showed a later eating time for the last meal (*p* < 0.001), longer eating duration (*p* < 0.001), and later caloric midpoint (*p* = 0.037) than day workers. Late eaters from both workgroups consumed more 24 h energy (*p* = 0.028), fat in calories (*p* = 0.006) and protein (calories: *p* < 0.001; percentage of total calories: *p* = 0.042), and less carbohydrates in calories (*p* = 0.010) intake than early eaters. The time of the first meal was negatively correlated with 24 h energy (*p* = 0.024) and carbohydrate (*p* = 0.031) intake only in day workers. The time of the last meal was positively correlated with 24 h energy (day workers: β = 0.352; *p* = 0.044; night workers: β = 0.424; *p* = 0.002) and protein (day workers: β = 0.451; *p* = 0.013; night workers: β = 0.536; *p* < 0.001) intake for both shift workers, and with carbohydrate (β = 0.346; *p* = 0.016) and fat (β = 0.286; *p* = 0.042) intake only in night workers. Eating duration was positively correlated with energy (day workers: β = 0.473; *p* = 0.004; night workers: β = 0.320; *p* = 0.023) and carbohydrate (day workers: β = 0.418; *p* = 0.011; night workers: β = 0.364; *p* = 0.010) intake in both groups. Thus, time-related eating patterns indicative of intake later at night are associated with increased daily energy and macronutrient intake.

## 1. Introduction

Shift work is a work schedule that covers the entire 24 h as a consequence of the needs of modern society [1]. The work hours are outside of the “standard” work day and rest at night, and work is performed at irregular hours, especially at night [2]. Due to working at atypical hours, shift workers usually experience insufficient sleep [3,4] and circadian misalignment [3,5]. Findings from studies also demonstrated a higher frequency of overweight/obesity [4,6] and other nutritional and metabolic diseases, such as dyslipidemias [7,8], metabolic syndrome [9,10], and type 2 diabetes [11,12,13], among shift workers.

The consumption of a poor diet has been associated with the development of nutritional disorders in shift workers [14,15,16]. In addition, the time when meals are usually eaten has recently been associated with metabolic problems in shift workers [17,18,19,20,21], including police officers [20,21], as well as in the general population [22,23]. In this sense, extending the feeding window into the night with late meals and reducing the nightly fasting period seems to contribute to poor nutritional health since the metabolism of nutrients is less efficient at night compared to the daytime period [24,25,26]. Thus, increased calorie intake at night could have negative metabolic consequences [27].

The study of dietary patterns related to the time of day has brought a new approach to nutritional studies, especially due to the demonstrated relationship between such patterns and metabolic diseases [28,29,30], as well as the overall quantity and quality of food eaten [31,32,33], including food with high carbohydrate, fat, and sugar content. These variables include the time of meals, daily distribution of energy and nutrients, the 24 h food window (eating duration), and caloric midpoint, which is the time of day when 50% of the calories consumed in the day are reached [34]. Although research on the subject has shown important associations between temporal eating patterns and nutritional diseases [29,35], as well as poorer nutritional quality [36], it is largely unknown whether the time of eating impacts the energy and macronutrient intake of shift workers. Considering that energy balance is a crucial point in the development, prevention, and treatment of obesity [37,38], the aim of the present study was to investigate whether time-related eating patterns are associated with the daily intake of calories and macronutrients in the day and night workers. Our hypothesis was that dietary patterns indicative of higher nighttime intake (higher eating duration, later time of the first and the last meal, higher caloric midpoint, higher nighttime caloric intake) are associated with higher energy and macronutrients intake in day and night shift workers.

## 2. Materials and Methods

### 2.1. Participants and Ethics

This cross-sectional study was conducted with 81 male shift workers (29 day shift workers and 52 night shift workers) made up of military police officers from a medium-sized Brazilian city between February 2019 and February 2020. Day shift workers started working between 06:00 h and 08:00 h and finished between 15:00 h and 17:00 h. Night shift workers started working between 21:00 h and 23:00 h and finished between 06:00 h and 07:00 h. Participants included in the study worked in the administrative area or worked outside on the streets doing patrols. Eligible participants were males between the age of 20 and 50 and working on shift for at least one year. Participants with type 2 diabetes, uncontrolled hypertension, sleep disorders, and related mood disorders and depression were excluded. All the participants provided written informed consent. The study was approved by the Ethics Committee of the Federal University of Uberlândia (CAAE: 68216417.5.0000.5152).

Sample size was estimated using G*Power software version 3.1.9.2 (Heinrich Heine University Düsseldorf, Düsseldorf, Germany). An F-test for linear regression (fixed model, R2 deviation from zero) was performed with a statistical power (1–β err prob) of 0.80, a medium effect size of 0.30, and an overall level of significance of 0.05. The minimum sample size of 29 participants was obtained for each group (day and night shift workers).

### 2.2. Initial Assessment

A questionnaire created by the research team was applied to the participants and included personal information regarding sociodemographic characteristics such as age, marital status, education, sleep habits; lifestyle, including smoking, alcohol consumption, physical exercise; medications; food intake; and weight changes after starting the current shift. Information related to the health problems of the participants was obtained by the occupational health sector of the military police.

### 2.3. Instruments

#### 2.3.1. Anthropometric Evaluation

Weight and height were measured according to Lohman et al.’s standards [39]. Weight was measured with a set of scales to an accuracy of 0.1 kg (Welmy W300) and height with a stadiometer fixed to the wall, with an accuracy of 0.1 cm (Welmy W300). Both measurements were used to calculate the body mass index (BMI, kg/m^2^). Waist circumference (WC) was measured at the level of the umbilicus using an inextensible anthropometric tape (Sanny Medical, SN-4010, precision of 0.5 cm).

#### 2.3.2. Food Intake Evaluation

All the participants, day and night shift workers, answered a dietary record of three non-consecutive days. In the present study, as workdays often coincided with weekend days, the participants were instructed to fill in the food recall for two days of work and one day off. Food and fluid intake, including brand names and recipes for home-cooked foods, were provided by the participants in as much detail as possible. Portion sizes were estimated using individual food items/units in addition to common household measurements such as cups, glasses, bowls, teaspoons, and tablespoons. When participants returned the forms, they discussed with a qualified nutritionist to include additional explanations and details to improve the accuracy of the information obtained. Energy and nutrients such as carbohydrate, fat, and protein intake were analyzed and performed using Dietpro version Clinic 5.8 software. Data were analyzed for total intake and intake per kg of body weight.

#### 2.3.3. Time-Related Eating Patterns

Time-related eating patterns, including the number of meals, eating duration, the time of the first and last meal, 12 h nightly fasting, and the caloric midpoint, were calculated by the research team from the 3-day food recall—two days of work and one day of rest. This methodology has been widely applied in chrononutritional studies [33,40,41]. First, the number of meals was established by the number of caloric events ≥50 kcal/day with time intervals of ≥15 min between meals [42] reported in the dietary record. The eating duration was calculated as a difference in hours between the first and the last meal consumed in the day [34], and the night fasting interval was obtained by calculating the longest fasting interval between eating episodes from 19:00 to 07:00 [33]. The distribution of energy and macronutrients throughout the day of day and night workers was analyzed according to periods: Period 1—05:00–10:59; Period 2—11:00–16:59; Period 3—17:00–22:59; Period 4—23:00–04:59. These intervals were determined according to the concentration of the largest number of participants who had meals in the respective time intervals, considering day and night workers. The caloric midpoint was calculated as the time at which 50% of each individual’s daily calories were consumed [6], and the mean of the three days of dietary record was calculated. Then, the median of the caloric midpoint—an important marker of the probability of an extended and later eating window [6]—of the participants was used to classify the day and night workers as “early” (≤15:50 h) or “late” (>15:50 h) [40] eaters to compare the total energy and macronutrient intake between the two groups of shift workers.

#### 2.3.4. Sleep Assessment

A wrist actigraphy monitor (ActTrust, Condor Instruments^®^, São Paulo, Brazil) was used to assess the sleep–wake pattern and was validated for shift workers [43]. The participants used the actigraphy monitor for seven consecutive days, including work days and days off. Participants also filled in a 7day sleep diary [44] on the same week of the food evaluation. The software ActStudio (Condor Instruments^®^—version 1.0.0.0050.2015) was used to analyze the data and obtain the mean sleep duration.

#### 2.3.5. Chronotype and Social Jetlag

Chronotype was assessed using the Munich Chronotype Questionnaire (MCTQ) derived based on the time of mid-sleep time on free days (MSF), with a further correction for sleep debt—calculated as the difference between average sleep duration on the work days and days off. Social jetlag was calculated as the absolute difference between the time of mid-sleep on work days and days off [45].

### 2.4. Statistical Analysis

All statistical analyses were performed using SPSS version 21.0 (IBM Corp, Armonk, NY, USA).

#### 2.4.1. Initial Analysis

Generalized Linear Models (GzLM) with sequential Sidak post hoc were used to compare day and night workers according to the following characteristics: sociodemographic and lifestyle (dependent variables). The work shift was included in the model to compare day and night shift workers. The qualitative variables were expressed as the frequency in percentage, and absolute number and the comparison between shifts was performed using the chi-square analysis.

#### 2.4.2. Comparisons of Time-Related Eating Patterns between Day and Night Workers

Generalized Estimating Equations (GEE) with sequential Sidak post hoc were used to compare the differences between day and night workers and time-related eating patterns, which were included as dependent variables. The crude data obtained from the 3-day food recall were used to perform the statistical analysis of food intake and time-related eating patterns. The exposure variable was the work shift, and the outcome was daily calorie/macronutrient intake and the time-related eating pattern variables (number of meals, the time of the first and the last meal, eating duration, 12-h nightly fasting, time of the caloric midpoint). The GEE analysis was adjusted for age and BMI.

The comparison of energy and macronutrient intake (dependent variables) between early and late eaters in different work shifts was also performed by GEE. The model was constructed as follows: isolated effects of the shift and caloric midpoint groups (late and early) and the interaction between them. The post hoc sequential Sidak was used. The results were represented as mean ± standard error of the mean (SEM). All analysis was adjusted for age and BMI.

#### 2.4.3. Analysis of the Distribution of Energy and Macronutrients throughout the Day of Day and Night Workers According to Periods

The GEE was also used to analyze the interaction between shift and the distribution of energy and macronutrient intake (the dependent variables) during specific time windows, with sequential Sidak post hoc. The isolated effect of shift and time window and the interaction between them were obtained. The results were represented as mean ± standard error of the mean (SEM) or mean and confidence interval. The GEE analysis was adjusted for age and BMI.

#### 2.4.4. Association Analysis between Time-Related Eating and the Daily Intake of Calories and Macronutrients

Multiple linear regression modeling analysis separated by work shift, adjusted for age and BMI, was used to analyze the association between time-related eating patterns and total daily intake. Independent variables were eating duration, 12 h nightly fasting, number of meals, caloric midpoint, social jetlag, and sleep. Dependent variables were energy, carbohydrates, fat, protein, and total food consumption. Work days and days off were grouped together for all analyses, as there were no significant differences between them (previous analyses). Statistical tests with *p* < 0.05 were accepted as significant.

## 3. Results

Table 1 describes the sociodemographic characteristics, lifestyle, anthropometric, and sleep variables comparing day and night male shift workers. Night workers had been working shifts for longer than day workers. A greater proportion of night workers practiced physical exercise regularly compared to day workers. Social jetlag was higher in night workers than day workers, whereas sleep duration was higher in day workers than night workers.

Compared to day shift workers, night shift workers presented a later time of last meal, longer eating duration, shorter nightly fast, and a later caloric midpoint (Table 2). Night workers also had higher protein intake (both in calories and in calories per kg of body weight) compared with day workers. No significant differences were found for the other nutrients in calories/macronutrients per kg of body weight between shifts.

Energy and macronutrient distribution in time windows according to the work shift is shown in Figure 1. Day workers stopped their food intake earlier than night workers, who usually had an extended eating window (window 4). There was no significant effect of the interaction between shift and the time windows (1, 2, and 3) for energy, carbohydrate, fat, and protein. A significant effect of the period was found for energy and all macronutrients (*p* < 0.001 for all). Individual values of energy/macronutrients and the effect of the eating period are presented in the Appendix A.

The total daily energy and macronutrient intake of early and late eaters according to the shift are shown in Table 3. There was no significant effect of the interaction between the caloric midpoint and shift for energy, carbohydrate, fat, and protein intake. An effect of shift showed that day workers consumed a higher percentage of energy than night workers. A significant effect of the caloric midpoint showed that late eaters consumed more energy, fat, and protein compared to early eaters. Early eaters consumed more carbohydrates compared to late eaters regardless of shift work. No significant differences were found in the analysis of the interaction between shift and caloric midpoint for calories/macronutrients per kg.

Table 4 shows that the number of meals was positively associated with energy and carbohydrate intake in both groups of shift workers and only with protein intake in day workers. The time of the first meal was negatively associated with energy and carbohydrate intake only in day workers. The time of the last meal was positively associated with energy and protein intake for both groups of shift workers and with carbohydrate and fat intake only in night workers. The eating duration was positively associated with energy and carbohydrate intake in both groups of shift workers and with fat and protein intake only in day workers. The nightly fast duration was negatively associated with energy, carbohydrate, and protein intake in the day and night workers. The caloric midpoint was positively associated with energy, fat, and protein intake only in the day workers. Lastly, social jetlag was positively associated with fat intake only in the night workers.

## 4. Discussion

In the present study, it was investigated whether time-related eating patterns are associated with the daily intake of calories and macronutrients in day and night male shift workers. In general, time-related eating patterns indicative of late-night intake seem to be associated with increased daily energy and macronutrient intake. These results confirm our hypothesis that eating late is associated with a greater intake of energy in both groups of shift workers (night and day) and highlight that the control of daily caloric intake in shift workers must include the accurate assessment and management of eating times.

Results from the present study showed that night shift workers stopped consuming food later compared to day workers. This eating pattern commonly observed in night workers [16,46] explains the other results found in the present study, such as the later time of the last meal, the greater eating duration, and the later caloric midpoint. Furthermore, the positive association found between the eating duration and energy and carbohydrate intake in both groups of individuals suggests that if the shift worker sleeps less at night, it seems that he uses part of that time to eat. In the present study, a significant difference was found between both groups of shift workers for the time of the last meal but not for the time of the first meal, which is likely due to night workers finishing their shift in the morning and having breakfast.

Evidence shows that night-shift workers generally have a poor diet [14,47,48], and our results highlight that the time of eating seems to be a route that predisposes not only the night worker but also the day worker to have this worse eating pattern in terms of quantity of energy and macronutrients. In this sense, a positive caloric balance during the 24 h period of the day due to the excessive caloric intake at night could predispose them to the risk of obesity [49] and metabolic disorders [50]. In the general population, night consumption was shown to increase the daily food intake and contribute to the development of obesity [51,52]. A study conducted by Peplonska, Kaluzny, and Trafalska [16] showed that shift workers who eat more at night tend to have higher rates of obesity. As we found in the present study, other studies performed with police officers also showed later meal times on night shift days [21]. Interestingly, findings from studies have also shown that insufficient sleep is associated with poorer quality of diet consumed by police officers [20].

The results found among day workers in the present study showed, as occurred with night workers that a longer eating duration or later timing of the last meal seems to be associated with a higher total energy intake. However, even with some associations also found in day workers, it is important to emphasize that night workers have a routine imposed by the work schedule that contributes to irregular meal times throughout the day, including the night period, which is more easily manageable among day workers. In this study, day workers possibly presented some associations between time-related eating patterns and total daily calories due to personal preference and/or other activities/obligations performed on the day, not necessarily due to the work schedule as in the case of night workers.

In our study, workers from both shifts who ate later consumed more energy and total fat and protein than those who ate earlier. These results showed that, regardless of the work shift, a late eating pattern is associated with a higher total daily intake of energy and macronutrients. These results corroborate data from other studies that also found that eating at night is associated with increased daily energy intake but also changes the consumption of macronutrients [51,53,54].

Given our results, it is important to reflect on why workers consume food at a time when they are not physiologically prepared for food intake. According to studies on this topic, the habit of eating at night in shift workers is due to a series of factors capable of altering the daily quantity and quality ingested. Environmental factors such as lack of places to eat, limited healthy food choices, unavailability of time, easy access to unhealthy food, and the influence of attitudes and preferences of colleagues can affect the food consumption of night workers [15]. This may worsen the quality of food and increase the number of calories due to the increase in the consumption of discretionary food, which is one of the factors that contribute to the development of metabolic disorders [15]. Furthermore, later meals are associated with high total energy intake, which can often deteriorate the daily dietary pattern [53,55,56]. In addition, consuming a high energy proportion at night seems to be inappropriate in terms of metabolic response compared to the morning response [27,57]. The impact of nighttime eating promoting an unfavorable metabolic response could affect weight regulation [27,58]. Furthermore, night eating has been associated with impaired work performance and attention [59] and decreased satiety function [60,61] and may result in metabolic disruption and obesity and increase the risk of several diseases, such as type 2 diabetes [12,62,63], metabolic syndrome [9,10], and cardiovascular diseases [11,64].

Results from the present study showed that social jetlag was positively correlated with total fat consumption in night workers. Findings from studies by our group also showed associations between social jetlag and daily food intake [65,66,67], such that a higher degree of social jetlag is associated with a later meal time (especially at breakfast ); greater total daily calorie, carbohydrate, and fat intake; and higher intake of sweets. The measure of social jetlag, which is related to the discrepancy between circadian and social clocks, was also associated with obesity [68,69,70] and its related diseases [66,70]. This could be one factor contributing to poor nutritional health found in the present study.

The present study has some limitations. The cross-sectional design limits the establishment of a cause-and-effect relationship. Although previously validated in other studies, evaluations using subjective questionnaires are dependent on the participants’ memory and motivation. Lastly, only male police officers were studied, and the generalization of the results to other settings, occupations, and women cannot be made.

## 5. Conclusions

We conclude that regardless of work shift time-related eating patterns, which indicate later time of the last meal, greater eating duration, and later caloric midpoint, are associated with an increase in daily energy and macronutrient intake in shift workers. Further studies are necessary to better understand the long-term associations between time-related eating patterns and daily food consumption in shift workers, especially to know the potential of this dietary pattern leading to the development of metabolic diseases such as obesity.

## Figures and Tables

**Figure 1 nutrients-14-02202-f001:**
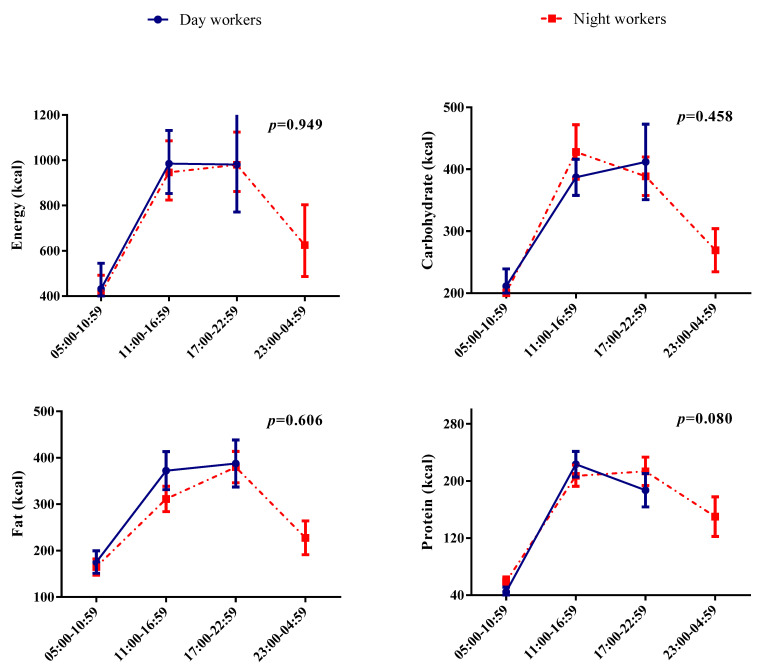
Energy and macronutrients distributed in periods according to the shift. Data represented as mean and confidence interval. Generalized Estimated Equations (GEE) were used to analyze the interaction (*p*-value in the figure) between shifts and the distribution of energy and macronutrients in periods. Sequential Sidak post hoc was used, and the analysis was adjusted for age and BMI. *p*-interaction values < 0.05 were accepted as significant. Catarina–cc.

**Table 1 nutrients-14-02202-t001:** Sociodemographic characteristics, lifestyle, anthropometric variables, and sleep variables of participants according to the work shift.

	Day Shift Workers (*n* = 29)	Night Shift Workers (*n* = 52)	*p*
Age (years)	36.4 ± 0.9	38.5 ± 0.7	0.081
Marital status			
Single (%)	20.7 (6)	25.0 (13)	0.661
Married (%)	79.3 (23)	75.0 (39)	
Schooling			
High-school (%)	24.1 (7)	28.8 (15)	0.650
Graduate (%)	62.1 (18)	63.5 (33)	
Postgraduate (%)	13.8 (4)	7.7 (4)	
Smoking			
Yes (%)	3.4 (1)	15.4 (8)	0.101
No (%)	96.6 (28)	84.6 (44)	
Alcoholic beverages			
Yes (%)	62.1 (18)	75.0 (39)	0.222
No (%)	37.9 (11)	25.0 (13)	
Regular physical exercise			
Yes (%)	51.7 (15)	90.4 (47)	<0.001
No (%)	48.3 (14)	9.6 (5)	
Time of shift work (years)	3.4 ± 0.5	7.1 ± 0.8	<0.001
Weight (kg)	91.6 ± 2.1	87.7 ± 1.5	0.131
Height (m)	1.78 ± 0.01	1.76 ± 0.00	0.086
WC (cm)	98.5 ± 1.6	95.9 ± 1.2	0.201
BMI (kg/m²)	28.7 ± 0.6	28.2 ± 0.4	0.519
Sleep duration (hours)	5.9 ± 0.2	4.8 ± 0.1	0.002
MSF	3:20 ± 0:20	3:40 ± 0:15	0.443
Social jetlag (hours)	1.2 ± 0.1	6.1 ± 0.4	<0.001

BMI: body mass index; WC: waist circumference; MSF: mid-sleep time on free days. Data represented as mean ± standard error of the mean (SEM). Generalized Linear Model (GLzM) was used to compare day and night shift workers. *p*-values < 0.05 were accepted as significant.

**Table 2 nutrients-14-02202-t002:** Time-related eating patterns, 24 h energy, and macronutrient intake between day and night shift workers.

	Day Shift Workers (*n* = 29)	Night Shift Workers (*n* = 52)	*p*
Number of meals	3.9 ± 0.1	4.1 ± 0.1	0.185
Time of the first meal (h)	8.4 ± 0.3	8.8 ± 0.2	0.288
Time of the last meal (h)	20.5 ± 0.2	23.1 ± 0.2	<0.001
Eating duration (hours)	12.1 ± 0.4	14.4 ± 0.3	<0.001
Night fasting (hours)	10.0 ± 0.4	7.3 ± 0.2	<0.001
Caloric midpoint (h)	14.8 ± 0.4	15.9 ± 0.3	0.037
Energy (kcal)	2329.2 ± 198.3	2774.9 ± 170.4	0.095
Energy (kcal/kg)	24.8 ± 2.0	27.9 ± 1.5	0.184
Carbohydrate (kcal)	980.7 ± 89.6	1193.7 ± 91.1	0.099
Carbohydrate (kcal/kg)	31.1 ± 3.3	36.0 ± 2.5	0.175
Fat (kcal)	903.6 ± 88.1	1005.5 ± 71.1	0.375
Fat (kcal/kg)	9.6 ± 0.8	10.0 ± 0.6	0.691
Protein (kcal)	444.9 ± 39.7	575.6 ± 37.8	0.020
Protein (kcal/kg)	4.7 ± 0.3	5.8 ± 0.3	0.038

Data represented as mean ± standard error of the mean (SEM). Generalized Estimated Equations (GEE) were used to compare the time-related patterns and food consumption between day and night shift workers, adjusted for age and BMI. *p*-values < 0.05 were accepted as significant.

**Table 3 nutrients-14-02202-t003:** Comparison of daily energy and macronutrient consumption of early eaters and late eaters according to the shift.

Energy and Macronutrients	Day Workers (*n* = 29)		Night Workers (*n* = 52)		Shift	Caloric Midpoint	Shift and Caloric Midpoint Interaction
	Early Eaters (*n* = 19)	Late Eaters (*n* = 10)	Early Eaters (*n* = 18)	Late Eaters (*n* = 25)	*p*	*p*	*p*
Energy (kcal)	2024.7 ± 135.0	2907.9 ± 462.5	2611.3 ± 242.3	2894.9 ± 233.5	0.238	0.028	0.222
Energy (kcal/kg)	22.1 ± 1.6	30.0 ± 3.3	26.7 ± 2.5	28.8 ± 2.1	0.405	0.035	0.204
Energy (%)	35.4 ± 1.0	35.7 ± 1.2	31.4 ± 0.9	32.1 ± 0.7	<0.001	0.597	0.816
Carbohydrate (kcal)	307.5 ± 23.2	405.8 ± 70.8	324.7 ± 30.7	340.5 ± 34.2	0.606	0.167	0.328
Carbohydrate (kcal/kg)	29.1 ± 2.5	35.0 ± 4.5	36.6 ± 3.8	35.6 ± 4.0	0.268	0.470	0.338
Carbohydrate (%)	44.2 ± 1.7	39.7 ± 2.5	46.8 ± 1.9	40.1 ± 2.1	0.525	0.010	0.629
Fat (kcal)	264.0 ± 21.8	418.7 ± 65.9	258.5 ± 30.5	302.4 ± 23.0	0.126	0.006	0.180
Fat (kcal/kg)	8.2 ± 0.7	12.3 ± 1.4	9.5 ± 1.17	10.4 ± 0.7	0.938	0.019	0.143
Fat (%)	36.5 ± 1.5	40.3 ± 2.0	34.0 ± 1.7	36.4 ± 1.4	0.062	0.071	0.758
Protein (kcal)	131.4 ± 8.4	204.1 ± 29.3	133.3 ± 10.4	187.3 ± 16.5	0.715	<0.001	0.613
Protein (kcal/kg)	4.1 ± 0.2	6.0 ± 0.8	4.9 ± 0.4	6.5 ± 0.5	0.181	0.001	0.560
Protein (%)	19.1 ± 0.7	19.8 ± 1.5	19.1 ± 1.0	23.5 ± 1.4	0.165	0.042	0.151

Data represented as mean ± standard error of the mean (SEM). Generalized Estimated Equations (GEE) were used to analyze the interaction between shift and caloric midpoint. Sequential Sidak post hoc was used, and the analyses were adjusted for age and BMI. *p* values < 0.05 were accepted as significant.

**Table 4 nutrients-14-02202-t004:** Association between time-related eating patterns and daily food consumption between day and night shift workers.

Independent Variables	Dependent Variables															
	Day Shift Workers (*n* = 29)								Night Shift Workers (*n* = 52)							
Time-Related Eating Patterns	Energy (kcal)		Carbohydrate (kcal)		Fat (kcal)		Protein (kcal)		Energy (kcal)		Carbohydrate (kcal)		Fat (kcal)		Protein (kcal)	
	β	*p*	β	*p*	β	*p*	β	*p*	β	*p*	β	*p*	β	*p*	β	*p*
Number of meals	0.406	0.022	0.470	0.005	0.250	0.189	0.409	0.031	0.363	0.011	0.414	0.004	0.227	0.113	0.213	0.144
Time of the first meal	−0.401	0.024	−0.374	0.031	−0.365	0.051	−0.351	0.068	0.080	0.572	−0.055	0.705	0.127	0.364	0.255	0.068
Time of the last meal	0.352	0.044	0.299	0.082	0.284	0.124	0.451	0.013	0.424	0.002	0.346	0.016	0.286	0.042	0.536	<0.001
Eating duration	0.473	0.004	0.418	0.011	0.412	0.020	0.505	0.004	0.320	0.023	0.364	0.010	0.158	0.266	0.267	0.060
Night fasting	−0.396	0.019	−0.356	0.032	−0.284	0.119	−0.546	0.002	−0.446	0.001	−0.408	0.004	−0.270	0.052	−0.520	<0.001
Caloric midpoint	0.421	0.014	0.243	0.162	0.475	0.007	0.501	0.005	0.102	0.482	0.027	0.854	0.068	0.633	0.265	0.063
Social jetlag	−0.040	0.827	−0.101	0.576	−0.016	0.932	0.314	0.756	0.209	0.151	0.166	0.265	0.282	0.047	0.011	0.942
Sleep duration	−0.266	0.144	−0.171	0.340	−0.332	0.077	−0.204	0.298	0.074	0.652	0.155	0.347	−0.010	0.951	−0.031	0.850

Multiple Linear Regression modeling analysis was used to identify the association between time-related eating patterns. social jetlag/sleep duration and food consumption (energy and macronutrients) separated by work shift and adjusted for age and body mass index. *p* values < 0.05 were accepted as significant.

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
