# Peer review of "Time-Related Eating Patterns Are Associated with the Total Daily Intake of Calories and Macronutrients in Day and Night Shift Workers"

_nutrients, 2022, doi:10.3390/nu14112202_

Round 1

Reviewer 1 Report

The work entitled “Time-related eating patterns are associated with the total daily intake of calories and macronutrients in day and night shift workers” analyses the time-related eating patterns associated with the daily intake of calories and macronutrients of Brazilian military police officers. It was seen that high late-night intake leads to increased energy and macronutrient intake. The work is well structured and organized. The methodology employed is clearly described. The data is scientifically sound and, even though there are some English writing mistakes, the discussion is well done and supported by literature. In general, the work is extremely methodic and up to date, dealing with a very important issue.

I would only recommend authors to revise the work again for English mistakes.

Author Response

Comments to the Author

The work entitled “Time-related eating patterns are associated with the total daily intake of calories and macronutrients in day and night shift workers” analyses the time-related eating patterns associated with the daily intake of calories and macronutrients of Brazilian military police officers. It was seen that high late-night intake leads to increased energy and macronutrient intake. The work is well structured and organized. The methodology employed is clearly described. The data is scientifically sound and, even though there are some English writing mistakes, the discussion is well done and supported by literature. In general, the work is extremely methodic and up to date, dealing with a very important issue.

I would only recommend authors to revise the work again for English mistakes.

Response: We thank the reviewer for reviewing our manuscript. The manuscript underwent further detailed revision following the English grammar rules.

Reviewer 2 Report

This well-written manuscript is an interesting manuscript on the association between the timing and the content of food intake in shift workers. In my view it reads well and the findings are interesting to the readership of Nutrients.

I have a few comments and I hope that they will be helpful to the authors.

  1. The 3-day food recall was performed on non-consecutive days. How were these specific days chosen: By the participants or the study investigators? It would have been better to have consecutive days/nights followed by one day off. For better reproducibility, were the 3 recorded days of the same shift type between all participants? If not, this would introduce heterogeneity and lack of comparability between participants. In Table 2, were the food intake and macronutrient content the same between working days/nights vs. rest days?

1b. On a related note, once the food recall data were analyzed for energy and macronutrient content, did the authors also consider the intake of fibers and alcohol? Did participants report other information on eating behavior, snacking, a self-reported eating window or sleep times? Energy and macronutrient content could be also reported in kcal/kg (Table 2+3 and related text) and the statistical tests done accordingly.

  1. This cross-sectional study was conducted in police officers working day or night shifts, however it is not entirely clear whether they were chronically or new on this specific shift schedule, or if there was a change of shifts just a few days/weeks before the data collection (also relates to comment no.1). Other studies have shown different food intake and eating behavior between rotating shift workers and night shift-only workers. Presumably, those who joined a new police force unit/section recently with its own schedule scheme would have changed their food intake as well.

  1. The sample size calculation seems straightforward, however the parameters to perform the calculation and to reach a n=81 are unclear. Most often the sample size is estimated depending on the research question and the hypothesized changes/differences of the primary outcome, but in this study it is unclear what the research question was, what the primary outcome was, and what difference of this outcome was expected between day and night workers. Moreover, did the sample size calculation warrant the asymmetrical group size ratio (n=29 day vs. n=52 night shift workers, which is not exactly a 1:2 ratio)? Or was this a convenience sample?

  1. The Methods section defines the 4 periods (lines 117-118), 5h-11h, 11h-17h, 17h-23h and 23h-5h. How were these time intervals determined? Based on other studies or preliminary data? Other reasons to split time-related eating patterns into 4 intervals, instead of 6, 8, 10 or 12 periods?

  1. Probably a semantic issue. A statistical association cannot be positive or negative, it is either significant or not depending on the p-value and the chosen cutoff p-value. I suspect the authors meant that the correlation or coefficient was either positive or negative once they have checked for statistical significance. Please edit the phrases “positively associated” and “negatively associated” throughout the text to “positively correlated” and “negatively correlated”, respectively.

Minor comments

- Abstract, line 15: “mean of age” could read “mean age”.

- Abstract + Results: The sex of the studied population should be stated more clearly in the abstract and results section at least, because weight and eating behavior can vary with the menopausal status and menstrual cycles in some women.

- Introduction, lines 38-41: Please consider simplifying the multiple “and[s]” in this sentence.

- Methods, line 81: I presume the authors meant a “t-test”, not a “test-t”.

- Methods, lines 87-89: Were the questionnaires standardized and validated in this population? If so, please specify which questionnaire or score were used.

- Methods, line 144: Replace “percentual” by “percentage”.

- Results, line 175: The number is presumably a p-value, right?

- Results, line 184: Replace “stop” by “stopped”.

- Figure 1. To help the readers, please add the clock times of each period below the x-axis. The figure legend is hard to follow with so many numbers. The authors may consider moving these number results to a (supplementary?) table if relevant.

- Results, lines 202-213 + Table 3. Because the p-values of the text are also in Table 3, the text would be easier to follow with less numbers and the authors could refer the readers to Table 3 for specific p-values.

- Results, lines 220-236 + Table 4. Similar comment to Table 3. In addition, because of the numerous coefficients and p-values, the authors could present these results in a heatmap instead. Were the p-values corrected for multiple comparisons?

- Discussion, lines 250-252. I am not sure I follow the meaning of this last sentence. Please rephrase.

- Discussion, line 260. Good point. Was the quality of the consumed food assessed in this study? Line 270. The longer eating duration found in night workers is only due to a later last meal, but there was no difference in the time of first meal between day and night workers. Was this also the case in other studies? Please comment.

- Discussion, limitations paragraph. Please add that only male police officers were studied (if that’s correct). If the sample size was estimated correctly (comment no.3), then the n=81 participants is not a limitation and should be enough to answer the research question. On line 315, to be more precise, you could state “the generalization of the results to other settings, occupations and women cannot be done”, or something to that effect.

Author Response

Reviewer: 2

Comments to the Author

This well-written manuscript is an interesting manuscript on the association between the timing and the content of food intake in shift workers. In my view it reads well and the findings are interesting to the readership of Nutrients.

I have a few comments and I hope that they will be helpful to the authors.

Response: We thank the reviewer for the excellent considerations.

1a. The 3-day food recall was performed on non-consecutive days. How were these specific days chosen: By the participants or the study investigators? It would have been better to have consecutive days/nights followed by one day off. For better reproducibility, were the 3 recorded days of the same shift type between all participants? If not, this would introduce heterogeneity and lack of comparability between participants. In Table 2, were the food intake and macronutrient content the same between working days/nights vs. rest days?

Response: We thank the reviewer for this important consideration. Both day and night shift participants were instructed by the study investigators to complete a 3-day food food recall including two working days and one day off. The specific days of work and day off were relayed to the study investigators by the participants. The food recall was performed on non-consecutive days because it is already known that the diet of a given day can influence the dietary consumption of the following day (Willet, 2013). Thus, it is suggested to use non-consecutive days, covering weekdays and weekends. In addition, studies have shown that the assessment on non-consecutive days is more reliable to represent the usual food intake of individuals, which was our objective in the present study.

In the present study, as workdays often coincided with weekend days, the participants were instructed to fill in the food recall for two days of work and one day off. This happens in the same way for the two evaluated groups of shift workers and is better explained in the “Materials and Methods section”, “Food intake evaluation”:

“All the participants, day and night shift workers, answered a dietary record of three non-consecutive days. In the present study, as workdays often coincided with weekend days, the participants were instructed to fill in the food recall for two days of work and one day off. Food and fluid intake, including brand names and recipes for home-cooked foods, were provided by the participants in as much detail as possible. Portion sizes were estimated using individual food items/units in addition to common household measurements such as cups, glasses, bowls, teaspoons, and tablespoons. When participants returned the forms, they discussed with a qualified nutritionist to include additional explanations and details improving the accuracy of the information obtained. Energy and nutrients such as carbohydrate, fat and protein consumption were analyzed and performed using Dietpro version Clinic 5.8 software.  Data were analyzed for total intake and intake per kg of body weight.”

Before the analysis in Table 2, the comparison of work day and day off consumption was performed and there was no significant difference between them. The calories and macronutrient consumption were the same comparing work days and days off.

Reference

Willet WC. Nature of variation in diet. In: Nutritional epidemiology. 2 ed. New York: Oxford University Press, p. 34-48, 2013.

1b. On a related note, once the food recall data were analyzed for energy and macronutrient content, did the authors also consider the intake of fibers and alcohol? Did participants report other information on eating behavior, snacking, a self-reported eating window or sleep times? Energy and macronutrients content could be also reported in kcal/kg (Table 2 + 3 and related text) and the statistical tests done accordingly.

Response: We thank the reviewer. As suggested, we included the energy and macronutrients content reported in kcal/kg of body weight in Tables 2 and 3 and related text in the “Results” section.

Although this is an interesting approach, we did not include fiber and alcohol consumption in this study because it was not our main objective. At that moment, we wanted to evaluate the temporal consumption of energy and macronutrients. We appreciate your interesting suggestion and hope to be able to do so soon.

We also emphasize that all the information on eating behavior was extracted from the 3-day food recall and calculated by the research team. This methodology has been widely applied in chrononutritional studies (Adafer et al. 2020, Garcez et al. 2021, Gontijo et al. 2019, Marot et al. 2021, Teixeira et al. 2019).

“Time-related eating patterns, including the number of meals, eating duration, the time of the first and last meal, 12-h nightly fasting, and the caloric midpoint, were calculated by the research team from the 3-day food recall – two days of work and one day of rest. This methodology has been widely applied in chrononutritional studies [33,40,41]  First, the number of meals was established by the number of caloric events ≥50kcal/day with time intervals of ≥15min between meals[42] reported in the dietary record. The eating duration was calculated as a difference in hours between the first and the last meal consumed in the day[34] and the night fasting interval was obtained by calculating the longest fasting interval between eating episodes from 19:00 to 07:00.[33]

The sleep times were obtained from the 7d sleep diary and the wrist monitor actigraphy for seven days, including work days and days off. This information is available in the “Materials and Methods” section, “Sleep assessment” and Table 1.

“A wrist actigraphy monitor (ActTrust, Condor Instruments®) was used to assess the sleep-wake pattern and was validated for shift workers[43]. The participants used the actigraphy monitor for seven consecutive days, including work days and days off. Participants also filled in a 7d sleep diary[44] in the same week of the food  evaluation.”

References:

Adafer R et al. Food timing, circadian rhythm and chrononutrition: a systematic review of time-restricted eating’s effects on human health. Nutrients. 2020, 12, doi: 10.3390/nu12123770.

Garcez MR et al. A chrononutrition perspective of diet quality and eating behaviors of Brazilian adolescents in associated with sleep duration. Chronobiol Int. 2021, 38, 387-399, doi: 10.1080/07420528.2020.1851704.

Gontijo CA et al. Time-related eating patterns and chronotype are associated with diet quality in pregnant women. Chronobiol Int. 2019, 36, 75-84, doi: 10.1080/07420528.2018.1518328.

Marot LP et al. Eating duration throughout a rotating shift schedule: a case study. J Am Coll Nutr. 2021, 40, 624-631, doi: 10.1080/07315724.2020.1814899.

Teixeira GP et al. The association between chronotype, food craving and weight gain in pregnant women. J Hum Nutr Diet. 2020, 33, 342-350, doi: 10.1111/jhn.12723. 

  1. This cross-section study was conducted in police officers working day or night shifts; however it is not entirely clear whether they were chronically or new on this specific shift schedule, or it there was a change of shits just a few days/weeks before the data collection (also relates to comment no.1). Other studies have shown different food intake and eating behavior between rotating shift workers and night shift-only workers. Presumably, those who joined a new police force unit/section recently with its own schedule scheme would have changed their food intake as well.

Response: We thank the reviewer for this important observation. One of the inclusion criteria of the present study for both day and night shift workers was that they had to have worked in the respective shift for at least one year. Thus, in the case of shift change days before data collection, the participant was not included in the study. This information is detailed in “Materials and Methods” section, “Participants and Ethics”, paragraph 1, as follows:

“Eligible participants were male between the age of 20 and 50 and working in shift for at last one year.”

We also describe the average time working in the current shift in Table 1 (Time of shift work - years: 3.4±0.5 for day workers; 7.1±0.8 for night workers). Thus, we can say that shift workers are chronically exposed to the work routine.

We have highlighted all this information for your observation.

  1. The sample size calculation seems straightforward, however the parameters to perform the calculation and to reach a n=81 are unclear. Most often the sample size is estimated depending on the research question and the hypothesized changes/differences of the primary outcome, but in this study it is unclear what the research question was, what the primary outcome was, and what difference of this outcome was expected between day and night workers. Moreover, did the sample size calculation warrant the asymmetrical group size ratio (n=29 day vs. n=52 night shift workers, which is not exactly a 1:2 ratio)? Or was this a convenience sample?

Response: We thank the reviewer for this important consideration, which allowed us to notice a mistake in the information related to the sample size calculation. This study is part of a larger study and we made a mistake entering data from another article. We apologize for that.

First, it is important to emphasize that this study was conducted with a convenience sample and we highlight this as a limitation of the study (“Discussion” section, paragraph 5). Our primary outcome is total daily energy intake. Based on this, our hypothesis was that dietary patterns indicative of higher nighttime consumption are associated with higher energy and macronutrient intake both in day and night shift workers. Thus, our main research question was whether a higher nighttime consumption (higher eating duration, later time of the first and the last meal, higher caloric midpoint, higher nighttime caloric consumption) is associated with higher daily energy intake in both day and night shift workers. It is important to note that we did this separately within each group in the linear regression analyses, so that it was possible to discuss whether the associations between the exposure and outcome variables were different between the two groups. Considering that the temporal patterns of eating of night and day workers are very different and also expected, we concluded, after numerous attempts at analysis, that we should do this separately within each group.

Based on your consideration and due to the nature of the sample (convenience), it might not be necessary to present the sample size calculation. However, we did this to ensure that there was a minimum number of participants for this analysis within each group. Thus, the correct sample calculation in G power was performed as follows and

this information is described in the section “Materials and Methods” – subheading “Participants and Ethics”, paragraph 2.

“Sample size was estimated using G*Power software version 3.1.9.2 (Heinrich-Heine-University Düsseldorf, Düsseldorf, Germany). A F-test for linear regression (fixed model, R2 deviation from zero) was performed with a statistical power (1 – β err prob) of 0.80, a medium effect size of 0.30, and an overall level of significance of 0.05. The minimum sample size of 29 participants was obtained for each group (day and night shift workers).”

Unlike most studies (Bucher Della Torre et al. 2020, Lauren et al. 2020, Lin et al. 2020), we evaluated a greater number of night workers, which may make the sample size safer to answer the research question. It is noteworthy that night workers are more likely to have health problems related to working hours compared to day workers (Brum et al. 2020, Dutheil et al. 2020; Lim et al. 2018). As can be seen throughout the text of the article, these individuals are the main focus of the present study.

Thank you again for this important contribution to our article.

References:

Bucher Della Torre S et al. Energy, nutrient and food intakes of male shift workers vary according to the schedule type but not the number of nights worked. Nutrients. 2020, 12, doi: 10.3390/nu12040919.

Brum MCB et al. Night shift work, short sleep and obesity. Diabetology & metabolic syndrome. 2020, 12, doi:10.1186/s13098-020-0524-9.

Dutheil F et al. Shift work and particularly permanent night shifts, promote dyslipidaemia: A systematic review and meta-analysis. Atherosclerosis. 2020, 313, 156-169, doi:10.1016/j.atherosclerosis.2020.08.015.

Lauren S et al. Free-living sleep, food intake, and physical activity in night and morning shift workers. J Am Coll Nutr. 2020, 39, 450-456, doi: 10.1080/07315724.2019.1691954.

Lim YC et al. Association between night-shift work, sleep quality and metabolic syndrome. Occup Environ Med. 2018, 75, 716-723, doi: 10.1136/oemed-2018-105104. 

Ting-Ti L et al. Shift work relationships with same- and subsequent-day empty calorie food and beverage consumption. Scand J Work Environ Health. 2020, 46, 579-588, doi: 10.5271/sjweh.3903.

  1. The Methods section defines the 4 periods (lines 117-118), 5h-11h, 11h-17h, 17h-23h and 23h-5h. How were these time intervals determined? Based on other studies or preliminary data? Other reasons to split time-related eating patterns into 4 intervals, instead of 6, 8, 10 or 12 periods?

Response: We thank the reviewer for this interesting question. This stage is always very laborious and challenging in studies in the area of nutrition. As individuals are shift workers, they have a very different eating schedule than usual and the in most cases the determination of time intervals cannot be based on those in other studies/ general population. Therefore, we decided that a deep analysis of the distribution of consumption of this specific population would be the best way to do it. In addition, there are numerous types of shifts, a huge variation of working hours throughout the week and individuals who differ in their food consumption pattern, which affect the determination of time intervals. A given categorization may work well for one group, but it may be totally inadequate to describe the food windows of others.

For all these reasons, after hard work and many attempts to categorize these intervals, these were the ones that best described the food consumption of the workers in the present study. Intervals were determined according to the concentration of the largest number of participants who had meals in the respective time intervals, considering day and night workers. The intervals were split in four, following the different periods of the day (morning, afternoon, night and late-night). Other studies have divided the food consumption in periods across the day (Grant et al. 2021; Chamorro et al. 2022). This information is explained in the “Materials and Methods” section, “Time-related eating patterns” as follows:

The distribution of energy and macronutrients throughout the day of day and night workers was analyzed according to periods: Period 1 – 05:00-10:59; Period 2 – 11:00-16:59; Period 3 – 17:00-22:59; Period 4 – 23:00-04:59. These intervals were determined according to the concentration of the largest number of participants who had meals in the respective time intervals, considering day and night workers.”

References:

Grant LK et al. Time-of-day and meal size effects on clinical lipid markers. J Clin Endocrinol Metab. 2021, 106, doi: 10.1210/clinem/dgaa739.

Chamorro R et al. Meal timing across the day modulates daily energy intake in adult patients with type 2 diabetes. Eur J Clin Nutr. 2022, doi: 10.1038/s41430-022-01128-z.

  1. Probably a semantic issue. A statistical association cannot be positive or negative, it is either significant or not depending on the p-value and the chosen cutoff p-value. I suspect the authors meant that the correlation or coefficient was either positive or negative once they have checked for statistical significance. Please edit the phrases “positively associated” and “negatively associated” throughout the text to “positively correlated” and “negatively correlated”, respectively.

Response: We thank the reviewer for this consideration. We have corrected the terms referred to throughout the “Abstract”, “Results” and “Discussion” sections.

Minor comments

- Abstract, line 15: “mean of age” could read “mean age”.

Response: We thank the reviewer. The correction has been made in the “Abstract” section.

- Abstract + Results: The sex of the studied population should be stated more clearly in the abstract and results section at least because weight and eating behavior can vary with the menopausal status and menstrual cycles in some women.

Response: We thank the reviewer for this observation. The present study included only male military police officers. This information was included in the “Abstract”, “Materials and Methods” – “Participants and Ethics” (paragraph 1) –, “Results” (paragraph 1) and “Discussion” (paragraph 1) sections.

- Introduction, lines 38-41: Please consider simplifying the multiple “and[s]” in this sentence.

Response: We thank the reviewer. The correction has been made in the “Introduction” section, paragraph 1.

- Methods, line 81: I presume the authors meant a “t-test”, not a “test-t”.

Response: We thank the reviewer. The correction has been made in the “Materials and Methods” section – “Participants and Ethics”, paragraph 2.  

- Methods, lines 87-89: Were the questionnaires standardized and validated in this population? If so, please specify which questionnaire or score were used.

Response: We thank the reviewer. The questionnaire of the initial assessment was created by the research team. As it is a tool that aimed to evaluate general and simple information from the participants, which was not directly related to the research question, we considered that validation would not be necessary. This issue was better explained in the “Materials and Methods” section – “Initial assessment”:

“A questionnaire created by the research team was applied to the participants and included personal information regarding sociodemographic characteristics such as age, marital status, education, sleep habits; lifestyle, including smoking, alcohol consumption, physical exercise; medications; food consumption, and weight changes after starting the current shift. Information related to the health problems of the participants was obtained by the occupational health sector of the Military Police.”

 - Methods, line 144: Replace “percentual” by “percentage”.

Response: We thank the reviewer. The correction was made in the “Materials and Methods” section – “Statistical Analysis”, subheading “General data analysis”. 

- Results, line 175: The number is presumably a p-value, right?

Response: We thank the reviewer. The number is presumably a p-value.

- Results, line 184: Replace “stop” by “stopped”.

Response: We thank the reviewer for this observation. The correction was made in the “Results” section, paragraph 3.

- Figure 1. To help the readers, please add the clock times of each period below the x-axis. The figure legend is hard to follow with so many numbers. The authors may consider moving these number results to a (supplementary?) table if relevant.

Response: We thank the reviewer for this observation. The clock times of each period were added below the x-axis and in the Figure 1 legend. Numerical results from Figure 1 were moved to a Supplementary Material.

- Results, lines 202-213 + Table 3. Because the p-values of the text are also in Table 3, the text would be easier to follow with less numbers and the authors could refer the readers to Table 3 for specific p-values.

Response: We thank the reviewer for this suggestion. The numbers of results were removed from the text (“Results” section).

- Discussion, line 260. Good point. Was the quality of the consumed food assessed in this study? Line 270. The longer eating duration found in night workers is only due to a later last meal, but there was no difference in the time of the first meal between day and night workers. Was this also the case in other studies? Please comment.

Response: We thank the reviewer for this interesting consideration. The assessment of diet quality was not our objective in the present study. At this point, we focus on studying only the quantitative aspects of food intake.

Night shift workers had a greater time of last meal due to de work schedule, which explains the greater eating duration compared to day workers. However, as night workers finished the schedule in the morning, the time of the first meal was the same between day and night shift workers. Thus, the breakfast at the end of the work shift coincides with the breakfast time of day workers. That is why we did not find a significant difference between the time of the first meal between day and night shift workers. To improve understanding of this important point, we have included information about it in the “Discussion” section (paragraph 2).

- Discussion, limitations paragraph. Please add that only male police officers were studied (if that's correct). If the sample size was estimated correctly (comment no.3), then the n=81 participants are not a limitation and should be enough to answer the research question. On line 315, to be more precise, you could state "the generalization of the results to other settings, occupations and women cannot be done", or something to that effect.

Response: We thank the reviewer for the consideration. Your suggestions have been accepted and the text modified (“Discussion” section, paragraph 5) as follows:

“The present study has limitations. The cross-sectional design limits the establishment of a causal and effect relationship. Although previously validated in other studies, evaluations using subjective questionnaires and dependent on the participants’ memory and motivation are also limited. Lastly, only male police officers were studied and the generalization of the results to other settings, occupations and women cannot be done.”

- Results, lines 220-236 + Table 4. Similar comment to Table 3. In addition, because of the numerous coefficients and p-values, the authors could present these results in a heatmap instead. Were the p-values corrected for multiple comparisons?

Response: We appreciate your suggestion. We opted to remove the numbers from the text, according to your previous suggestion. As we have several time-related eating patterns (exposure variables) described in Table 4 (Number of meals, Time of the first meal, Time of the last meal, Eating duration, 12-h nightly fasting, etc.), we understand that the viewing these numbers (beta and significance values) can help the reader to understand the magnitude of these associations. This is important as it is closely related to our research question. That's why we chose to leave these values only in the table.

We inform that p-values were corrected for multiple comparisons.

 Thank you again for your contribution.

- Discussion, lines 250-252. I am not sure I follow the meaning of this last sentence. Please rephrase.

Response: We thank the reviewer. This sentence was rephrased (“Discussion” section, paragraph 1) as follows:

“In general, time-related eating patterns indicative of late-night intake seem to lead to increased daily energy and macronutrient intake.”

Reviewer 3 Report

Introduction:

The literature on dietary patterns in shift-working population of police/military officers is missing. There is a lot of field studies you can cite (see Kosmadopoulus et al. 2020, or Velazquez-Kronen et al. 2021).

Material and Methods:

Line 73: Much more details on the day shift workers and night shift workers are needed. Please provide more details on the study protocol. Did you study these workers on two consecutive work days and one day of rest? Please restructure the method section better. Please include subheading for Instruments, Data analysis ( with Time-related eating patterns subsection)

Line 79: Please provide the number of the ethical approval.

Line 125 (sleep assessment): It is not clear on which study days actigraphy was used. Which instruments were used on which study days? Did you assess sleep on the same days as the food evaluation (two days of work and then on the one day of rest)?

Statistical analysis: Please describe more detailed analyses that you have done. It is not sufficient only to write Generalized Linear Models. Which correlation structure in GLMs and GEE you have chosen for the dependencies between the study days?

It is not sufficiently clear which variables are exposure and which variables are outcome. Only from the tables does this become reasonably clear. Which confounding variables did you consider? And how did you select the confounding variables? You have workers with different job roles with different demands. Did you adjust or stratify for it? Which sensitivity analyses were done?

Table 4: did you put all exposure variables in one model, or is that one model for each exposure? Adjusted for which confounding variables? Please consider the paper on “Table 2 fallacy” https://pubmed.ncbi.nlm.nih.gov/23371353/

Figure 1: Please indicate that the y-axes are not starting at zero. Please indicate in the figure confidence intervals for the means instead of standard error. p-values in the figure: for which effect are they?

Discussion: Please do not cite any Tables or Figures in the discussion section. It would be nice to get the results presented in relation to the literature on police workers.

Author Response

Reviewer: 3

Introduction: The literature on dietary patterns in the shift-working population of police/military officers is missing. There is a lot of field studies you can cite (see Kosmadopoulus et al. 2020, or Velazquez-Kronen et al. 2021).

Response: We thank you for this relevant suggestion. We have cited both studies in the “Introduction” section, paragraph 2 and “Discussion” section, paragraph 3.

Material and Methods:

- Line 73: Much more details on the day shift workers and night shift workers are needed. Please provide more details on the study protocol. Did you study these workers on two consecutive workdays and one day of rest? Please restructure the method section better. Please include subheading for Instruments, Data analysis (with Time-related eating patterns subsection)

Response: We thank the reviewer. This information has been better explained in the “Materials and Methods” section, “Instruments”. Subheading “Instruments”, “Initial analysis”, “Comparison of time-related eating patterns between day and night workers”, “Analysis of the distribution of energy and macronutrients throughout the day of day and night workers according to periods”, Association analysis between time-related eating and the daily intake of calories and macronutrients” in the "Materials and Methods" section was created.

- Line 79: Please provide the number of the ethical approval.

Response: We thank the reviewer and apologize for the failure. The number of ethical approvals has been included in the “Materials and Methods” section, “Participants and Ethics”, paragraph 1.

- Line 125 (sleep assessment): It is not clear on which study days actigraphy was used. Which instruments were used on which study days? Did you assess sleep on the same days as the food evaluation (two days of work and then the one day of rest)?

Response: A wrist actigraphy monitor concomitant with a filled 7-d sleep diary was used to assess sleep data for seven consecutive days, including working day and day off in both, and in the same week of the food evaluation. This information was included in the “Materials and Methods” section, subheadings “Instruments” and “Sleep assessment” as follows:

“A wrist actigraphy monitor (ActTrust, Condor Instruments®) was used to assess the sleep-wake pattern and was validated for shift workers[43]. The participants used the actigraphy monitor for seven consecutive days, including workdays and days off. Participants also filled in a 7d sleep diary [44] in the same week of the food evaluation.”

- Statistical analysis: Please describe more detailed analyses that you have done. It is not sufficient only to write Generalized Linear Models. Which correlation structure in GLMs and GEE you have chosen for the dependencies between the study days?

Response: We thank the reviewer for this important consideration. We accepted your suggestion and changed the text, inserting more important information about what we have performed in the statistical analyses in the "Materials and Methods" section – "Statistical Analysis", as follows:

“Initial analysis”

“Generalized Linear Models (GzLM) with sequential Sidak post-hoc were used to compare day and night workers according to the following characteristics: sociodemographic and lifestyle (dependent variables). The work shift was included in the model to compare day and night shift workers. The qualitative variables were expressed as a frequency in percentage and absolute number and the comparison between shifts was performed using the chi-square analysis.”

“Comparisons of time-related eating patterns between day and night workers”

“Generalized Estimating Equations (GEE) with sequential Sidak post-hoc were used to compare the differences between day and night workers and time-related eating patterns, which were included as dependent variables. The crude data obtained from the 3-day food recall were used to perform the statistical analysis of food intake and time-related eating patterns. The exposure variable was the work shift and the outcome was daily calorie/macronutrient intake, and the time-related eating pattern variables (number of meals, the time of the first and the last meal, eating duration, 12-h nightly fasting, time of the caloric midpoint). The GEE analysis was adjusted for age and BMI.

The comparison of energy and macronutrient intake (dependent variables) between early and late eaters in different work shifts was also performed by GEE. The model was constructed as follows: isolated effects of the shift and caloric midpoint groups (late and early), and the interaction between them. The post-hoc sequential Sidak was used. Results were represented as mean ± standard error of the mean (SEM). All analysis was adjusted for age and BMI.”

“Analysis of the distribution of energy and macronutrients throughout the day of day and night workers according to periods”

“The GEE was also used to analyze the interaction between shift and the distribution of energy and macronutrient intake (the dependent variables) during specific time windows, with sequential Sidak post-hoc. The isolated effect of shift and time window and the interaction between them were obtained. Results were represented as mean ± standard error of the mean (SEM) or mean and confidence interval. The GEE analysis was adjusted for age and BMI.”

“Association analysis between time-related eating and the daily intake of calories and macronutrients”

“Multiple linear regression modeling analysis separated by work shift, adjusted for age and BMI, was used to analyze the association between time-related eating patterns and total daily intake. Independent variables were eating duration, 12-h nightly fasting, number of meals, caloric midpoint, social jetlag, and sleep. Dependent variables were energy, carbohydrates, fat, protein, and total food consumption. Workdays and days off were grouped together for all analyses as there were no significant differences between them (previous analyses). Statistical tests with p<0.05 were accepted as significant.”

- It is not sufficiently clear which variables are exposure and which variables are the outcome. Only from the tables does this become reasonably clear. Which confounding variables did you consider? And how did you select the confounding variables? You have workers with different job roles with different demands. Did you adjust or stratify for it? Which sensitivity analyses were done?

Response: As in the above consideration, we accepted your suggestion and changed the text, inserting more important information about what we have performed in the "Statistical Analysis", "Data analysis with time-related eating patterns", and in the "Materials and Methods" section.

The confounding variables considered were age and BMI, which were selected considering the variation of food intake that is evidenced in different age and BMI values. The statistical analysis performed was not adjusted or stratified for a different job role with different demands. Regardless of the job function, the level of stress seems similar due to the particular workload of the profession. Sensitivity analyses were not performed in the present study.

- Table 4: did you put all exposure variables in one model, or is that one model for each exposure? Adjusted for which confounding variables? Please consider the paper on “Table 2 fallacy” https://pubmed.ncbi.nlm.nih.gov/23371353/

Response: We thank the reviewer. We inform that each exposure variable was included in a model with adjustments for age and BMI. Thanks for the suggestion to read this interesting article.

- Figure 1: Please indicate that the y-axes are not starting at zero. Please indicate in the figure confidence intervals for the means instead of standard error. p-values in the figure: for which effect are they?

Response: We thank the reviewer. Figure 1 was adjusted in the y-axes indicating that they are not starting at zero (fat and protein macronutrients), and the standard error was replaced by confidence intervals. Additional information about it was included in the "Materials and Methods" section, subheadings "Statistical analysis", and "Analysis of the distribution of energy and macronutrients throughout the day of day and night workers according to periods”.

We inform that the p-values indicated in Figure 1 are related to the interaction between shift and calories/macronutrient consumption, cited in the figure legend.

- Discussion: Please do not cite any Tables or Figures in the discussion section. It would be nice to get the results presented in relation to the literature on police workers.

Response: We thank the reviewer for this observation. The citation of Tables and Figures was excluded from the "Discussion" section.

The results of the present study were related to the results of the literature on police officers. This inclusion was made in the “Discussion” section, paragraph 3.
